# Vehicle Make and Model Recognition as an Open-Set Recognition Problem and New Class Discovery

Diana-Itzel Vázquez-Santiago, Héctor-Gabriel Acosta-Mesa * and Efrén Mezura-Montes

Artificial Intelligence Research Institute, Universidad Veracruzana, Veracruz 91097, Mexico; diana.v.s@hotmail.com (D.-I.V.-S.); emezura@uv.mx (E.M.-M.)
* Correspondence: heacosta@uv.mx

**Abstract:** One of the main limitations of traditional neural-network-based classifiers is the assumption that all query data are well represented within their training set. Unfortunately, in real-life scenarios, this is often not the case, and unknown class data may appear during testing, which drastically weakens the robustness of the algorithms. For this type of problem, open-set recognition (OSR) proposes a new approach where it is assumed that the world knowledge of algorithms is incomplete, so they must be prepared to detect and reject objects of unknown classes. However, the goal of this approach does not include the detection of new classes hidden within the rejected instances, which would be beneficial to increase the model's knowledge and classification capability, even after training. This paper proposes an OSR strategy with an extension for new class discovery aimed at vehicle make and model recognition. We use a neuroevolution technique and the contrastive loss function to design a domain-specific CNN that generates a consistent distribution of feature vectors belonging to the same class within the embedded space in terms of cosine similarity, maintaining this behavior in unknown classes, which serves as the main guide for a probabilistic model and a clustering algorithm to simultaneously detect objects of new classes and discover their classes. The results show that the presented strategy works effectively to address the VMMR problem as an OSR problem and furthermore is able to simultaneously recognize the new classes hidden within the rejected objects. OSR is focused on demonstrating its effectiveness with benchmark databases that are not domain-specific. VMMR is focused on improving its classification accuracy; however, since it is a real-world recognition problem, it should have strategies to deal with unknown data, which has not been extensively addressed and, to the best of our knowledge, has never been considered from an OSR perspective, so this work also contributes as a benchmark for future domain-specific OSR.

**Keywords:** open-set recognition; new class discovery; VMMR; CNN; contrastive loss; clustering; neuroevolution



## 1. Introduction

Automatic vehicle make and model recognition (VMMR) aims to offer innovative services to improve the efficiency and safety of transportation networks. These services include intelligent traffic analysis and management, electronic toll collection, emergency vehicle notifications, the automatic enforcement of traffic rules, etc. In recent years, several authors have proposed and implemented different approaches and techniques to present solutions to the various challenges of VMMR such as the similar appearance of different vehicle models [1,2], variations in the images due to weather conditions or resolution [3–5], recognition through different key points or regions [6,7], etc. However, most of these solutions are designed within a *closed-set* approach, where it is assumed that all query data are well represented by the training set, and therefore these solutions lack mechanisms to detect during testing when an input sample does not belong to any of the predefined classes. These unforeseen situations are very likely to happen in real-life scenarios and drastically weaken the robustness of the models.

Every day, we have more and more access to labeled data, which makes data-hungry algorithms such as classification algorithms that employ supervised learning improve their classification accuracy by having more training information. However, it is unrealistic to think that we will be able to train these algorithms to recognize any object that may be presented to them. In the specific case of this application domain, it is estimated that there are currently more than 3300 vehicle makes in the world, for which models have been added and removed from the market, modifying the design in each generation and producing different versions of the same vehicle, which has made it very difficult to have a database containing enough examples of all the existing vehicles in circulation to correctly train a model. This limitation is very common in real-world recognition/classification tasks such as VMMR and, in most cases, results in misclassified vehicles because the algorithms were not prepared to deal with objects of unknown (novel) classes.

To solve this problem, some strategies have been proposed, such as periodically retraining the algorithms, incorporating an incremental update mechanism [8,9], using zero-shot [10,11] or one-shot (few-shot) [12,13] learning, etc. Although these strategies provide models with greater flexibility or the possibility of eventually increasing their classification potential, they do not address the fundamental problem of recognizing a novel class during testing (*open-set* problem). Scheirer et al. were the first to describe a more realistic scenario in which new classes not seen in training appear in testing and require classifiers not only to accurately classify objects of known classes but also to effectively deal with classes not considered in the training set [14]. They formalized this problem as *open-set recognition* (OSR) and proposed a solution called 1-vs-Set machine, where the risk of labeling a sample as known if it is far from the known data (*open space*) is measured, and its objective is to minimize this risk (*open-space risk*) by rejecting queries that lie beyond the reasonable support of the known data.

OSR led to extensive research that mostly focused on more effectively limiting the *open-space risk* [15–18], and little research was developed around efficiently performing *open-set recognition* and simultaneously discovering new classes hidden in the rejected instances. Some of the proposed solutions employed incremental learning [19], transfer learning [20,21], or clustering [22,23]. Although they achieved good results, most of them present limitations such as the determination of the number of new classes in a later or separate event from the recognition of novel instances, or the use of examples of unknown classes during validation, pretraining, or retraining stages as a strategy to fine-tune their representations/parameters; however, in OSR, there is almost never information of unknown classes.

In the specific case of VMMR, few works have been proposed that, although not described within an OSR framework, have mechanisms to deal with new classes. One of these studies was conducted by Nazemi et al. [3] from an anomaly detection approach. Their base system is capable of classifying 50 specific vehicle models, to which they added an anomaly detection based on a confidence threshold to identify vehicles that do not belong to any of these 50 classes. The "anomalies" are further classified based on their dimensions within two new classes: "unknown heavy" and "unknown light". Another approach was proposed by Kezebou et al. [12], with a few-shot learning approach requiring between 1 and 20 images for the generation of new classes.

In this paper, we propose to approach VMMR as an OSR problem extended for new class discovery. Since the known classes are supported by numerous well-labeled examples, we can very effectively train an image classification algorithm that employs supervised learning like convolutional neural networks (CNNs), which are the most widely used tool for this task. While these networks cannot deal with the recognition of new classes, their ability to extract meaningful features can be exploited to design a mechanism that can detect objects of new classes based on the distribution of feature vectors in the embedded space that, when aggregated between feature extraction and classification, would adopt an OSR approach. However, feature vectors are usually of high dimensionality, their distribution is not always clear, and there is no assurance that the behavior will be maintained in

instances of unknown classes, which can complicate the representation and interpretation of the space to detect new classes. To tackle these problems, we propose to train a CNN with contrastive learning using the contrastive loss function during the training stage to reorganize the space where the feature vectors are mapped. Instead of separating the images with a hyperplane, the contrastive loss function brings similar images in near space (in terms of, e.g., Euclidean distance, cosine similarity, or some other metric) and moves dissimilar images away, generalizing this behavior on new unseen data.

Although there are CNN architectures such as VGG16, AlexNet, etc., that have achieved state-of-the-art results in the most well-known benchmarks such as ImageNet, CIFAR-100, etc., we propose a new CNN architecture designed from images of the application domain of this work (VMM) and the contrastive loss function using a neuroevolution technique to ensure consistent distribution of feature vectors within the embedded space, which serves as the main guide for a probabilistic model and a clustering algorithm that carry out the detection of objects of new classes and simultaneously discover their classes.

The remainder of this paper is organized as follows: Section 2 presents the related work. Section 3 describes the proposed methodology to approach VMMR as an OSR problem with an extension for new class discovery. This section also presents the proposed global scheme and delves deeper into each stage, detailing how the techniques of neuroevolution, contrastive loss function, the probabilistic model, and clustering are linked so as to achieve the general purpose. Section 4 details the tests performed, including the parameters and justifications for each test and the results obtained at each stage with their respective interpretation. Finally, the conclusions are drawn, and future work is discussed in Section 5.

## 2. Literature Review

### 2.1. Open-Set Recognition

OSR [14] introduced a more realistic scenario for real-world recognition/classification tasks, where new classes not seen during training appear at query time during testing. To deal with these unforeseen situations, OSR algorithms have to consider that their knowledge of the world is incomplete and formulate strategies to minimize the risk of considering an unknown instance as known. The authors of [14] formalized this risk as an *open-set risk* $(R_O)$ in a probabilistic formulation (Equation (1)) as the relative measure of positively labeled open space $O$ compared with the overall measure of positively labeled space $S_O$.

$$R_O(f) = \frac{\int_O f(x)dx}{\int_{S_O} f(x)dx'} \tag{1}$$

where $f$ denotes a measurable recognition function.

Numerous studies have been conducted to minimize the risk of open sets and more effectively reject objects of unknown classes [15–18], which is the main goal of OSR. However, in a more desirable context, an OSR should go further and discover the unknown classes hidden inside the rejected objects. Within this context, some authors have proposed the use of incremental learning [19], transfer learning [20,21], or clustering [22]. Bendale and Boult [19] extended the *open-set recognition* problem to open-world recognition (OWR) to jointly consider the OSR and incremental learning of new classes. They proposed that an effective OWR system must perform four tasks: detecting unknown objects, choosing which objects to label for addition to the model, labeling these objects, and updating the model. In their paper, they presented the NNO algorithm. However, the tasks they proposed are not automated in the NNO, they require human supervision for labeling, and the determination of the number of classes happens in a later event after the recognition of new instances. In [20], Wang et al. studied the OWR problem in more detail by incorporating transfer learning to transfer knowledge from old classes to new ones. However, they needed to retrain their model with the presence of samples of unknown classes, which is a limitation since, in an OSR context, information from unknown classes is almost never available. A similar knowledge transfer proposal was presented by Han et al. [21]; however, they have

the same limitation since their idea was to pretrain their model with images of known and unknown classes. Another interesting proposal was developed in [22] by some authors of [21], where they also took advantage of the knowledge transfer approach but used clustering. The main limitation of this work is that they determined the number of new classes in a separate event from the discovery of new instances, which, as in [19], can lead to suboptimal solutions.

To our knowledge, the most related work to ours, in terms of simultaneously discovering the objects of new classes and these classes themselves, is [23]. They introduced a collective/batch decision-strategy-based OSR framework (CD-OSR) by slightly modifying the hierarchical Dirichlet process (HDP). CD-OSR first involves a co-clustering process in the training phase to obtain the appropriate parameters. In the testing phase, each known class is modeled as a group using a Gaussian mixture model (GMM) with an unknown number of subclasses (one or more subclasses representing the same class can be obtained), and the entire test set (collective/batch) is treated in the same way. Then, all the groups are co-clustered under the HDP framework, and each one is labeled as one of the known classes or as unknown, depending on whether the subclass assigned to it is associated with a known class or not. Other works on OSR such as [24] also took advantage of Gaussian distributions to obtain discriminative representations of the data to detect unknowns and classify knowns.

Another proposal that may be related to our work was presented in [18], where the OSR problem was addressed within a transfer learning approach using contrastive learning to model the data. They also highlighted the importance of developing and testing OSR solutions with domain-specific databases to test their efficiency in dealing with real-world applications. Unfortunately, this solution only rejects objects of new classes and does not include the discovery of their classes.

### 2.2. Neuroevolution and Contrastive Learning

In the field of evolutionary computation (EC), a technique called neuroevolution (NE) emerged to optimize artificial neural networks (ANNs) at different levels. Its current overall process can be summarized as follows: A random population of individual networks is generated (with a neural coding), real networks are created from them, and the networks are evaluated with a function that measures the quality of the results (the fitness function). The networks with the highest fitness are selected, certain random changes are introduced to generate offspring from them, and a new population (generation) is selected. This process is repeated until a certain level of fitness or number of generations is reached.

NE has achieved excellent results in this optimization task and has rapidly advanced toward the optimization of CNN topologies [25–32]. A crucial point in the performance of NE algorithms is neural encodings, which contain the topology information of an ANN and therefore have a great impact on the complexity of the search space. So, in order to implement this technique in CNNs, NE was faced with the problem of designing neural encodings that could abstract the parameters of CNNs in order to deal with these highly complex architectures. There are two types of neural encodings that are commonly employed: direct and indirect. Among the proposals using an indirect coding framework, we find works such as [25–27], and in the case of works that used indirect encodings, we find proposals such as [28–30]. In recent years, researchers have started to study a "hybrid" neural coding, which combines elements of the encodings mentioned above to eliminate some of their limitations. These "hybrid" representations have proved to be very useful to distribute the CNN representations in different substructures, leading to improvement in the search [31–33]. The advantages and disadvantages of different encoding schemes, as well as important niches of opportunity for future research, were described in detail in [34].

Although NE algorithms have a strategy to determine how well individuals are meeting the criterion (or criteria) being optimized (the fitness function), CNNs have their own strategy to quantify how close their predictions are to the expected output (the loss function), for which cross-entropy or negative log-likelihood are some of the most

frequently used functions. However, the study of these functions has continued, and alternatives have been proposed that have achieved superior results. In these advances, the supervised contrastive loss function (SupCon) [35] was developed following the contrastive learning approach but in a supervised environment, which allowed it to maintain the principle of mapping examples into the embedded space of contrastive learning (distance is minimized in terms of Euclidean distance, cosine similarity, etc., between similar objects and maximized for dissimilar objects) but take advantage of labeled data.

Although there are marked differences between various versions of contrastive loss functions, the family of contrastive loss functions, in general, considers the following: For A set of $N$ randomly sampled sample/label pairs (batch), $\{x_k, y_k\}$ $k = 1 \ldots N$ is considered; the corresponding batch used for training (multiviewed batch) consists of $2N$ pairs, $\left\{\tilde{x}_\ell, \tilde{y}_\ell\right\}_{\ell=1\ldots2N}$, where $\tilde{x}_{2k}$ and $\tilde{x}_{2k-1}$ are two random augmentations ("views") of $x_k(k = 1 \ldots N)$ and $\tilde{y}_{2k-1} = \tilde{y}_{2k} = y_k$. Given the above, SupCon is calculated as follows:

$$\mathcal{L}_{in}^{sup} = \sum_{i \in I} \mathcal{L}_{in,i}^{sup} = \sum_{i \in I} -\log\left\{\frac{1}{|P(i)|} \sum_{p \in P(i)} \frac{\exp\left(z_i \cdot z_p / \tau\right)}{\sum_{a \in A(i)} \exp\left(z_i \cdot z_a / \tau\right)}\right\} \tag{2}$$

where $i \in I \equiv \{1 \ldots 2N\}$ is the index of an augmented sample (anchor), $A(i) \equiv I \setminus \{i\}$ is the set of all the indices of the samples different than $i$, $P(i) \equiv \left\{p \in A(i) : \tilde{y}_p = \tilde{y}_i\right\}$ is the set of all positive sample indices different than $i$, $|P(i)|$ is its cardinality, the $\bullet$ symbol denotes the inner (dot) product, and $\tau$ is a scalar temperature parameter. SupCon's formulation generalizes the SimCLR loss function [36] to an arbitrary number of positive examples to deal with scenarios in which labels are available so that it is known that more than one sample can belong to the same class.

### 3. Materials and Methods

This section describes the methodology proposed to approach VMMR as an OSR problem with an extension for new class discovery. Figure 1 shows the overall process of our proposal, and the following subsections describe the process in detail, covering the following objectives:

1.  Employ an NE algorithm and contrastive learning to design a domain-specific CNN that generates feature vectors spatially close in terms of cosine distance if the instances belong to the same class and distant if they belong to different classes, preserving this behavior in instances of unknown classes.
2.  Implement a mechanism between the feature extraction and classification sections of the CNN capable of detecting objects of unknown classes and simultaneously discovering their classes, taking the mapping of feature vectors, described in the previous objective, as the main guide.
3.  Run a series of tests using the test set that includes images of classes with which the CNN was designed and trained (known) and images of new classes (unknown) to test that the algorithm is able to detect objects of unknown classes and simultaneously discover their classes.
4.  Classify images of known classes with a classification accuracy above 90%.

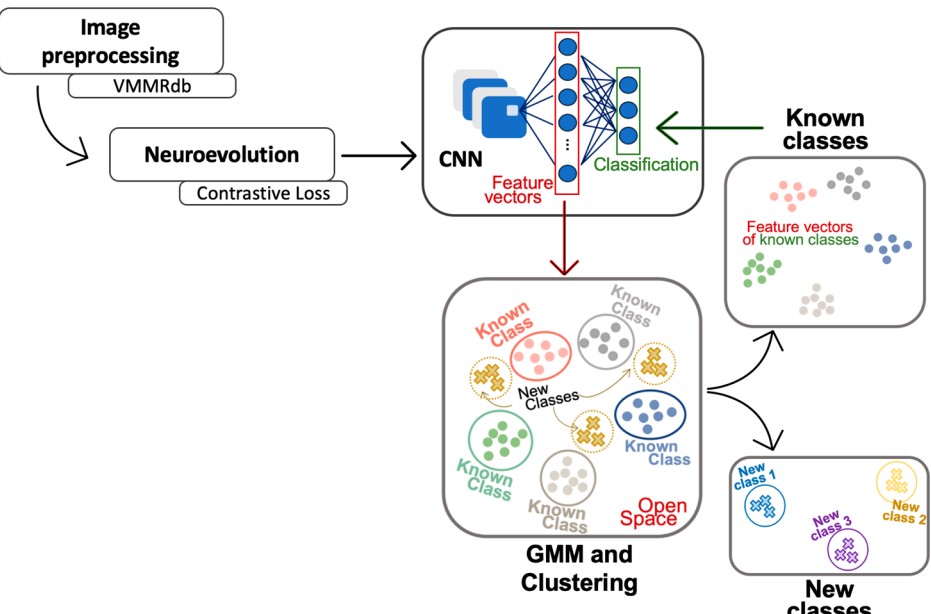

**Figure 1.** The proposed global process to approach VMMR as an OSR problem with an extension for new class discovery.

### 3.1. Dataset

The VMMRdb database [37] (available at https://github.com/faezetta/VMMRdb, accessed on 13 March 2023) was used in this work since it is one of the most cited in the specialized literature [12,38,39]. Only eight classes from the VMMRdb database were used and were manually filtered to retain only unduplicated images showing the rear view of the vehicles (i.e., samples of each class were not balanced). The filtered images were transformed to grayscale, resized to $28 \times 28$ pixels, and normalized with a mean of 0.456 and a standard deviation of 0.224.

Of the eight classes, five classes were used as "*known classes*": Chevrolet Silverado 2004, Ford Explorer 2002, Ford Mustang 2000, Honda Civic 2002, and Nissan Altima 2005. A sample of six images of each "*known class*" was used in the NE process to design the domain-specific CNN for VMMR. For the training of the resulting CNN from the NE process, the largest number of examples per class was needed, which had to be the same among different classes. However, due to the number of available samples in the database and the image filtering mentioned above, the final number of functional samples per "*known class*" varied between 75 and 250 images. Among the functional samples, three images of each class were kept for testing the complete OSR framework, and the rest were subjected to a data augmentation process to balance the number of examples per class, resulting in 250 images of each "*known class*". Furthermore, 200 images were used to train the CNN and model the "*known classes*" with a Gaussian mixture model (GMM), and the remaining 50 images were used to test the CNN classification accuracy and define the threshold of "*known classes*" in the GMM.

From the three remaining classes chosen from the database (Acura RSX 2003, Chevrolet Avalanche 2009, and Ford Escape 2011), three images of each class were chosen to only be used during the testing stage to represent "*unknown classes*" and validate that the proposed approach can detect them and discover their classes.

### 3.2. Neuroevolution and Contrastive Loss

One of the main objectives of this work is to exploit the ability of a CNN to extract meaningful features for designing a mechanism to detect objects of unknown classes based on the distribution of the feature vectors in the embedded space. To facilitate the interpretation of the embedded space, feature vectors extracted using the CNN are

considered to be spatially close in terms of cosine similarity if they belong to the same class and spatially distant if they belong to different classes, maintaining this behavior even if the classes are unknown. According to the state-of-the-art review, adding the contrastive loss function to the CNNs causes the feature vectors to be mapped in near space (in terms of, e.g., Euclidean distance, cosine similarity, or some other metric) if they are similar and far if they are dissimilar.

Although there are CNN architectures such as VGG16, AlexNet, etc., that have achieved state-of-the-art results in the most well-known benchmarks such as ImageNet, CIFAR-100, etc., we propose a new domain-specific architecture that would generate the previously described behavior in feature vectors, using the images mentioned in Section 3.1, an NE algorithm called DeepGA [33] (shown in Algorithm 1), and SupCon [35] expressed in Equation (2).

In [35], the authors made their PyTorch implementation of SupCon generally available (https://t.ly/supcon, accessed on 13 March 2023), and this was used in this work as a loss function in the CNNs generated in the NE process with DeepGA. (Originally, the negative log-likelihood loss was used.) The fitness function of DeepGA (Algorithm 1, line 15) was also modified to measure the desired behavior in feature vectors since optimization was the objective of our study. Thus, as the fitness function, we used the value of SupCon in the last training epoch of each generated CNN. Since the loss function decreases as the desired output is approached, DeepGA was set to work as a minimization problem, i.e., as the generated CNNs approached the desired target, the value of the loss/fitness function decreased.

The hybrid coding employed in DeepGA allows the algorithm to consider the number of fully connected layers and their corresponding number of neurons in its search for the best solution. However, during the NE process, it was detected that leaving the number of fully connected layers to DeepGA only increased the complexity and execution time since with only two fully connected layers, classification accuracies above 90% were achieved. To limit the number of fully connected layers during the evolutionary process, the first level of the mutation operator was modified. At the first level of the mutation operator, if $U_1(0,1) > 0.5$, a new block is added, and if $U_2(0,1) > 0.5$, the added block is a fully connected layer; then, the operator was modified so that if $U_2(0,1) > 0.5$, no block is added. This modification is shown in line 4 of Algorithm 2, which shows the mutation operator of DeepGA. This ensures that, during the whole evolutionary process, the generated networks only have two fully connected layers, allowing the algorithm's search to focus on the blocks of convolutional layers since they would be in charge of generating the feature vectors with the desired behavior.

To access the feature vectors generated using the CNNs, the *CNN class* of DeepGA, which builds the model for training and testing, was modified. As output, this class only generated the probabilities of the images belonging to the different classes. The modification consisted of the addition of the flattened outputs of the convolutional block (feature vectors) to the original output to be able to access them in the next process (i.e., to distinguish objects from new classes and simultaneously discover these classes).

The last modification to the DeepGA algorithm was an improvement in image reading. The PyTorch ImageFolder function was used to be able to read the images of all classes in a single process instead of reading the images of each class individually.

---

**Algorithm 1:** DeepGA pseudocode.

---

| | |
|---|---|
| **1** | **Input:** A population *P* of *N* individuals. The number of generations *T*, |
| **2** | crossover rate *CXPB*, mutation rate *MUPB*, tournament size *TSIZE*. |
| **3** | **Output:** |
| **4** | Initialize population (training the networks). |
| **5** | $t \leftarrow 1$ |
| **6** | **while** $t \leq T$ **do** |
| **7** | Select $N/2$ parents with probabilistic tournament selection |
| **8** | Offs $\leftarrow$ {} |
| **9** | **while** $|$Offs$| < N/2$ **do** |
| **10** | Select two random parents *p1* and *p2*. |
| **11** | **if** random(0,1) $\leq$ *CXPB* **then** |
| **12** | $O1, O2 \leftarrow$ Crossover(*p1*, *p2*) // Crossover |
| **13** | **if** random(0,1) $\leq$ *MUPB* **then** |
| **14** | Mutation(*O1*, *O2*) *// **Mutation (modified)*** |
| **15** | fitness(*O1*, *O2*) (Equation (1)) *// **Evaluation (modified)*** |
| **16** | $P \leftarrow P \cup$ Offs |
| **17** | Select the best *N* individuals in *P* as survivals. |
| **18** | **end** |
| **19** | **end** |

---

**Algorithm 2:** Mutation process DeepGA.

---

| | |
|---|---|
| **1** | **if** random(0,1) $\leq$ *MUPB* **then** |
| **2** | **if** random(0,1) $\leq$ U1 **then** // Adding a new block |
| **3** | **if** random(0,1) $\leq$ *U2* **then** |
| **4** | A convolutional block is added *// **Removed*** |
| **5** | **else** |
| **6** | A fully connected block is added |
| **7** | **else** // Restarting a block |
| **8** | **if** random(0,1) $\leq$ W1 **then** |
| **9** | Restarting a convolutional block |
| **10** | **else** |
| **11** | Restarting a fully connected block |

---

### 3.3. Neuroevolved CNN

The CNN architecture with the best fitness generated using DeepGA and SupCon was split to fulfill two purposes. First, the goal was to train the convolutional block with the contrastive loss function and the fully connected block with the cross-entropy loss function, using the full test set described in Section 3.1 (200 images of each of the five "*known classes*"), and to perform a classification accuracy test momentarily assuming a closed-set environment to validate that a good classification accuracy could be obtained since it is an essential point for OSR. Second, we sought to have the feature extraction process and the classification process separate since the detection of new class objects and the discovery of their classes must be accomplished between these events.

### 3.4. Gaussian Mixture Model (GMM) and Clustering

The main objective of this work is to approach VMMR as an OSR problem with an extension for the discovery of new classes. To achieve this, we divided our strategy into two phases, both relying on the consistent distribution of feature vectors in the embedded space generated using DeepGA and SupCon.

The first phase consisted of extracting the feature vectors from the images used to train the CNN and validating its classification accuracy. The feature vectors were compressed using principal component analysis (PCA) where the second and third components, which contributed 27.81 and 20.97 to the percentage of variance, respectively, were selected to perform a linear regression on the original feature vectors to obtain their projections. As

mentioned in Section 3.1, with the same proportion of data with which the CNN was trained and tested (80–20%), the 2D projections of the feature vectors were used to model each "*known class*" with a Gaussian mixture model (GMM) and define a recognition threshold of "*known classes*". In the test stage, where objects of both known and unknown classes were included, the GMM divided the objects as a group of unknown classes that did not pass the threshold and subsets of known classes whose probabilities matched the "*known class*" models. The above only served as a partial guide in the recognition of new class objects since, in the second phase of the strategy, a multiobjective clustering algorithm with automatic determination of the number of clusters (MOCKs) was employed and optimized with a multiobjective evolutionary algorithm (MOEA), called NSGA-II [40].

In the second phase, the clustering algorithm grouped the feature vectors extracted using the domain-specific CNN without any modification in their dimensionality. Since the GMM can determine the objects of known classes and their respective classes with some confidence, due to the threshold, we compared the subgroups of known classes generated using the GMM with the solutions of MOCK/NSGA-II to select the individual from the population with the highest similarity, where different criteria were used. First, the solutions that grouped the instances that the GMM determined as known and were in the same structures (subgroups) as the GMM had a higher score (one point for each shared structure). Although all the solutions of the clustering algorithm were optimal for the problem, we selected the solution that had the highest score (higher match with the GMM in the known classes) and was closest to the knee point as the "best solution".

Finally, we determined which clusters of the "best solution" contain known objects and separated them from the clusters containing unknown objects in a similar way to how the solutions were scored. Then, since the GMM also detected the objects of unknown classes (the objects that did not pass the threshold) with some confidence, those clusters that only contained objects that the GMM determined as unknown were automatically determined as new classes. After these processes, if there were still undetermined clusters as known or unknown, the number of known and unknown instances within the undetermined clusters were counted (according to the GMM determination), and the clusters were defined in the same category as that containing the majority of instances or as unknown if it contained the same number of examples to try to mitigate the *open-set risk*.

At the end of this strategy, the objects of the clusters that were determined as known were entered into the CNN's fully connected block to be classified, and the clusters that were determined to be unknown were the newly discovered classes of the objects detected as unknown.

The original version of the MOCK algorithm was proposed in 2004 by Handl et al. [41] and employed the MOEA called PESA-II. In 2016, Martinez-Peñaloza et al. [42] managed to improve the results by using the MOEA NSGA-II instead of PESA-II. In the MOCK version improved with NSGA-II, individuals are ranked and sorted according to their non-dominated level, and a crowding distance is used to perform niching. This distance is calculated for each member to be used by the selection operator to maintain a diverse front by ensuring that each member stays a crowding distance apart. Algorithm 3 shows NSGA-II's pseudocode.

**Algorithm 3:** NSGA-II pseudocode.

**1**   Initialize Population
**2**   Generate random population -size *M*
**3**   Evaluate Objective values
**4**   Assign Rank (level) Based on Pareto Dominance -"sort"
**5**   Generate Child Population
**6**   Binary Tournament Selection
**7**   Recombination and Mutation
**8**   **for** *i* = 1 to *Number of Generations* **do**
**9**       **for** *each Parent and Child in Population* **do**
**10**          Assign Rank (level) Based on Pareto –"sort"
**11**          Generate sets of non-dominated fronts
**12**          Loop (inside) by adding solutions to next generation starting
**13**          from the "first" front until *M* individuals found determine
**14**          crowding distance between points on each front
**15**      **end**
**16**      Select points (elitist) on the lower front (with lower rank) and
**17**      are outside a crowding distance. Create next generation
**18**      Binary Tournament Selection
**19**      Recombination and Mutation
**20 end**

## 4. Experiments and Results

This section describes the experiments and results obtained from our proposal to approach VMMR as an OSR problem with an extension for new class discovery.

For the neuroevolution process of CNNs performed with DeepGA [33] and Sup-Con [35], as mentioned in Section 3.1, six images of each "*known class*" taken from the VMMRdb [37] database were used.

The parameters described in Table 1 were used to initialize the population. Due to time constraints, it was not possible to use a parameter calibration program. Then, the parameters for the evolutionary process were calibrated manually. The parameters with which the best results were obtained, and which were used to generate the CNNs are shown in Table 2. The different values that each hyperparameter could have during the evolutionary process were the same as those established by the author of DeepGA and are presented in Table 3.

**Table 1.** Parameters for the population initialization.

| Parameter | Values |
|---|---|
| Min number of convolutional layers | 4 |
| Max number of convolutional layers | 9 |
| Min number of fully connected layers | 1 |
| Max number of fully connected layers | 1 |

**Table 2.** Parameters for the evolutionary process.

| Parameter | Values |
|---|---|
| Population size | 20 |
| No. of generations | 100 |
| Tournament size | 5 |
| Crossover rate | 0.7 |
| Mutation rate | 0.7 |
| No. of epochs per individual | 20 |

**Table 3.** Values that each hyperparameter could have during the evolutionary process.

| Hyperparameter | Values |
| --- | --- |
| No. of filters * | {2, 4, 8, 16, 32} |
| Filter size * | {2, 3, 4, 5, 6, 7, 8} |
| Pooling type * | {Max, Avg} |
| Pooling size * | {2, 3, 4, 5} |
| No. of neurons | {4, 8, 16, 32, 64, 128} |

Rows marked with * correspond to the convolutional block, while the unmarked row corresponds to the fully connected block.

Seven executions of the NE process with DeepGA and SupCon were performed. Figure 2 shows the convergence curves of the seven executions and a short analysis of the fitness values obtained in each one. As can be seen, all the executions started with a fitness within a range of 3.3 and 3.5, and most reached premature convergence or stalled at local optima. However, one of the executions (marked in red) achieved a more accurate search space that led to a fitness value of 1.96096.

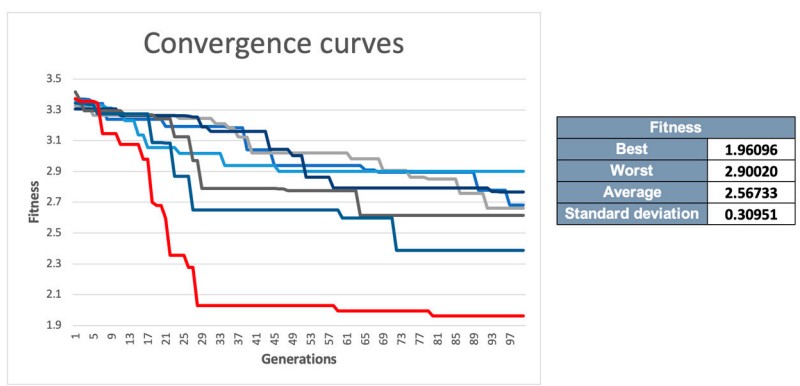

**Figure 2.** Convergence curves and the most relevant data of the fitness values of the seven runs of the NE process.

The CNN with the best fitness obtained in the NE process (henceforth referred to as the "domain-specific CNN") had a value of 1.96096, which was the value of SupCon in the last training epoch of the CNN (its justification is explained in Section 3.2 in more detail) and took 7 h to execute in the Visual Studio Code editor running on a MacBook Pro with a 2.2 GHz Quad-Core Intel Core i7 processor with 16 GB 1600 MHz DDR3 of memory. Figure 3 illustrates the architecture in terms of its encoding. In the first level, it can be observed that the architecture has 13 convolutional blocks (each one consisting of a single convolutional layer) and 2 fully connected blocks (each one comprising a single layer and a fully connected layer). The last convolutional block/layer generates feature vectors of 288 features. At the second level, the binary string defines the connectivity between convolutional blocks. Each bit represents the connectivity of a previous non-consecutive layer, starting from the third block. For a better understanding, we will explain three examples to understand the connections. The third convolutional block (first bit marked in red) can only have connections with previous blocks that are not its immediately previous consecutive block, so the third block cannot have a connection with the second convolutional block, but it can with the first one, which is why only one bit is assigned to it, and the bit value is 1. This means that there is a connection, which is represented by the red line on the first level. The next two bits (green) are for the fourth block, which can have a connection with the first or second block, and since the bit values are 1, both connections exist (represented by the green lines on the first level). A different case is shown in the next three bits (highlighted in yellow) assigned to the fifth block, which can have connections with the first, second, and third blocks; however, of those three bits, only the second one has a value of 1, which means that the fifth block only connects to the second block.

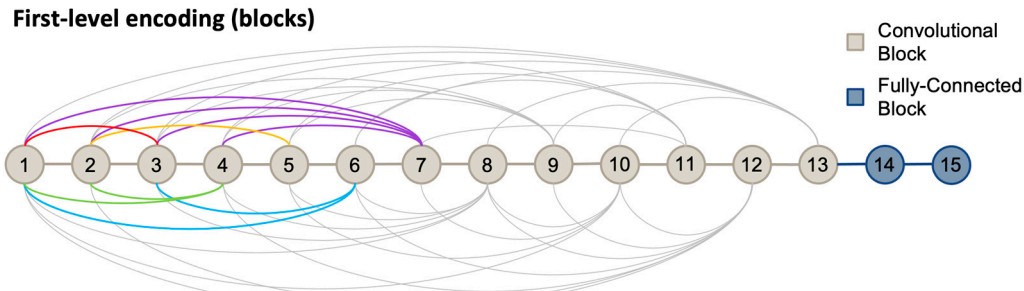

**Figure 3.** CNN architecture with the best fitness obtained using DeepGA and SupCon. The first level (blocks) represents simple convolutional operations instead of a set of convolutional layers. The second level (binary string) determines the skip connections received from the third block onward. Each bit represents the connectivity from previous layers, from the third layer onward.

To verify that the domain-specific CNN could generate the feature vectors extracted spatially close in terms of cosine similarity if they belong to the same class and far apart if they belong to different classes, a distance matrix using cosine similarity as the metric was generated with the feature vectors obtained in the last training epoch of the domain-specific CNN. On the same feature vectors, the t-SNE [43] technique was used to reduce the dimensionality from 288 to 2 in order to visualize them in a two-dimensional plane. The results of the distance matrix and t-SNE are shown in Figure 4. By means of these two techniques, it could be seen that the desired behavior in the feature vectors was achieved.

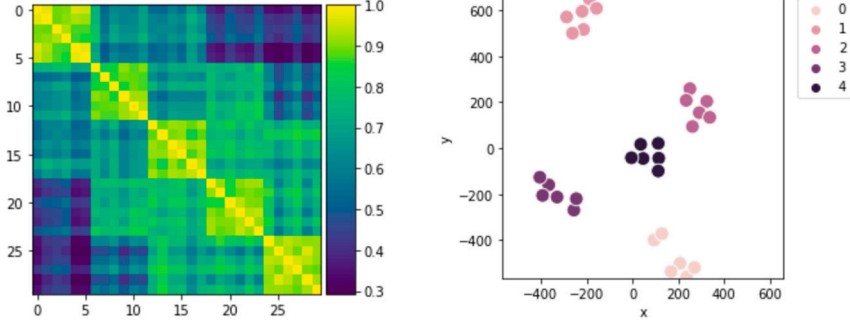

**Figure 4.** Distance matrix (with cosine similarity) and the projection in a two-dimensional plane of the feature vectors of the last training epoch of the domain-specific CNN.

As mentioned in Section 3.3, the domain-specific CNN was split to train the convolutional block with the contrastive loss function and the fully connected block with the cross-entropy loss function. For the training process, 1000 images of rear views of vehicles of the five "*known classes*" (200 images of each class) were used. A classification accuracy test was performed using 250 images of rear views of vehicles of the five "*known classes*" (50 images of each class) momentarily assuming a closed-set environment to validate that good classification accuracy was being achieved since it is an essential point for OSR. A 90% classification accuracy was reached during this test; more details regarding the data used are presented in Section 3.1.

The next test was to verify that the domain-specific CNN could generate feature vectors spatially close in terms of cosine similarity if they belong to the same class and far apart if they belong to different classes. This behavior was maintained in objects of unknown classes since the detection of objects of new classes and the discovery of their classes depended on this behavior. For this, the testing images, both the nine testing images of "*unknown classes*" shown in Figure 5 on the right and the fifteen images of the

five "*known classes*" shown in Figure 5 on the left, were entered into the convolutional block of the domain-specific CNN to extract their feature vectors. To visualize the results, which are shown in Figure 6, a distance matrix using cosine similarity as the metric was generated, and a two-dimensional projection was performed using linear regression with the two components described in Section 3.4. Figure 6 shows that the domain-specific CNN managed to generate feature vectors close in terms of cosine similarity if they belonged to the same class and distant if they belonged to different classes and managed to maintain such behavior even in objects of "*unknown classes*".

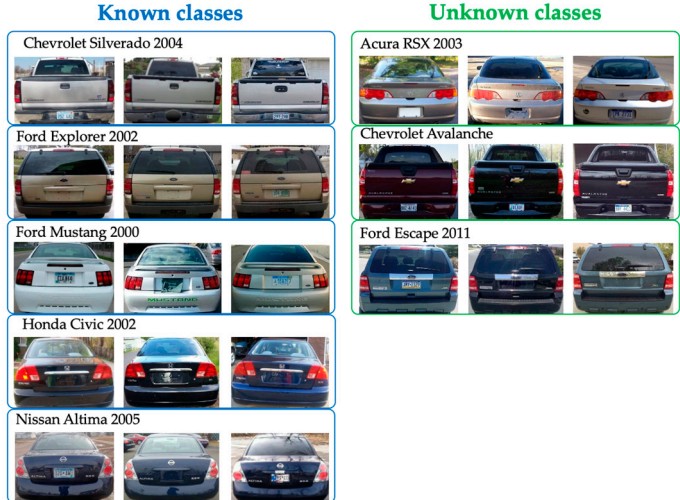

**Figure 5.** Sample images from the VMMRdb database. Both "*known*" (left) and "*unknown*" (right) classes were used during testing.

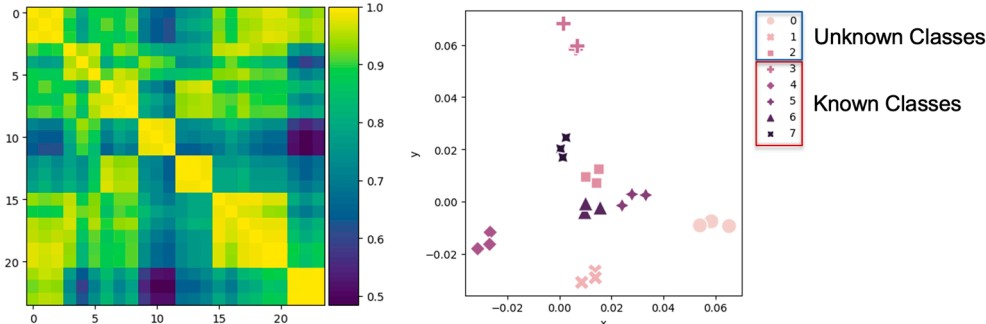

**Figure 6.** Distance matrix (with cosine similarity) and the projection in a two-dimensional plane of the feature vectors of the samples shown in Figure 5.

Later, the feature vectors of the images used to train the domain-specific CNN and validate its classification accuracy were compressed to two dimensions, and a linear regression was performed on these feature vectors to obtain their projections using the two components described in Section 3.4. With the projections of the 1000 images used to train the domain-specific CNN, we modeled the "*known Classes*" using a GMM, and the distribution of the Gaussians is shown in Figure 7. We then defined a "*known class*" recognition threshold within the GMM with a value of 9.999, using the projections of the 250 images that were used in the domain-specific CNN classification accuracy test.

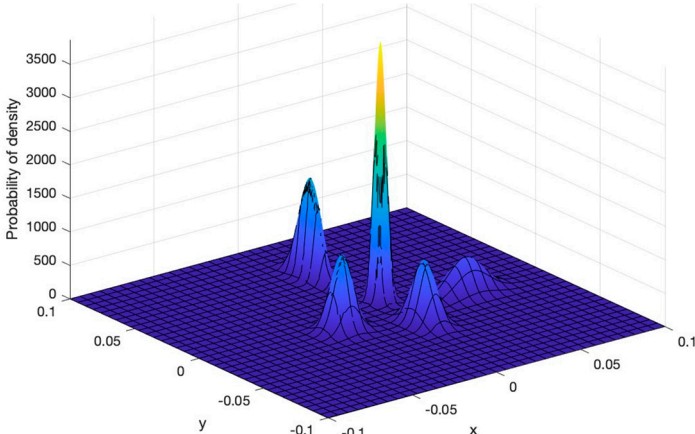

**Figure 7.** Distribution of the "known classes" using GMM.

Finally, the strategy proposed in Section 3.4 was carried out to detect the objects of new classes and discover their classes. The first step was to enter the two-dimensional projections used in Figure 6, which contain objects of both known and unknown classes (Figure 5), into the GMM to obtain their density probabilities. As its output, the model provided the probability of each object belonging to the known classes, and the threshold allowed us to set a probability limit for known or unknown classes. As can be seen in Table 4, the objects of known classes were correctly identified within their classes, and in the case of the objects of unknown classes, it can be seen that with the limit marked by the threshold, eight of the nine objects were correctly identified as unknown. A clearer representation of the results obtained can be seen in Figure 8, which indicates that the GMM divided the objects as a set of unknown classes that did not pass the threshold as well as the subsets of known classes whose probabilities matched the known class models.

**Table 4.** Probabilities of the test instances obtained with the GMM. The probabilities that exceeded the threshold (9999) are marked in orange.

| | x | y | Real Label | | Probability (Class 0) | Probability (Class 1) | Probability (Class 2) | Probability (Class 3) | Probability (Class 4) |
|---|---|---|---|---|---|---|---|---|---|
| 0 | $5.84 \times 10^{-2}$ | $-7.59 \times 10^{-3}$ | 0 | | $1.000 \times 10^{0}$ | $8.152 \times 10^{-27}$ | $7.294 \times 10^{-57}$ | $1.650 \times 10^{-89}$ | $1.483 \times 10^{-73}$ |
| 1 | $5.40 \times 10^{-2}$ | $-9.16 \times 10^{-3}$ | 0 | | $1.000 \times 10^{0}$ | $4.920 \times 10^{-22}$ | $6.332 \times 10^{-48}$ | $8.417 \times 10^{-77}$ | $1.826 \times 10^{-67}$ |
| 2 | $6.52 \times 10^{-2}$ | $-9.41 \times 10^{-3}$ | 0 | | $1.000 \times 10^{0}$ | $1.412 \times 10^{-33}$ | $3.304 \times 10^{-75}$ | $1.581 \times 10^{-112}$ | $1.321 \times 10^{-88}$ |
| 3 | $1.37 \times 10^{-2}$ | $-2.67 \times 10^{-2}$ | 1 | | $4.068 \times 10^{-17}$ | $1.000 \times 10^{0}$ | $6.451 \times 10^{-33}$ | $3.341 \times 10^{-22}$ | $8.823 \times 10^{-29}$ |
| 4 | $1.38 \times 10^{-2}$ | $-2.92 \times 10^{-2}$ | 1 | | $6.396 \times 10^{-19}$ | $1.000 \times 10^{0}$ | $1.408 \times 10^{-37}$ | $1.556 \times 10^{-23}$ | $2.039 \times 10^{-31}$ |
| 5 | $8.56 \times 10^{-3}$ | $-3.10 \times 10^{-2}$ | 1 | | $1.170 \times 10^{-22}$ | $1.000 \times 10^{0}$ | $3.308 \times 10^{-43}$ | $6.810 \times 10^{-24}$ | $3.034 \times 10^{-27}$ |
| 6 | $1.43 \times 10^{-2}$ | $7.13 \times 10^{-3}$ | 2 | Known | $1.020 \times 10^{-4}$ | $3.865 \times 10^{-6}$ | $9.999 \times 10^{-1}$ | $4.394 \times 10^{-9}$ | $2.252 \times 10^{-11}$ |
| 7 | $1.52 \times 10^{-2}$ | $1.24 \times 10^{-2}$ | 2 | | $1.451 \times 10^{-4}$ | $9.982 \times 10^{-9}$ | $9.999 \times 10^{-1}$ | $7.854 \times 10^{-8}$ | $1.672 \times 10^{-11}$ |
| 8 | $1.02 \times 10^{-2}$ | $9.51 \times 10^{-3}$ | 2 | | $5.701 \times 10^{-5}$ | $4.896 \times 10^{-6}$ | $9.999 \times 10^{-1}$ | $5.240 \times 10^{-7}$ | $1.290 \times 10^{-8}$ |
| 9 | $1.53 \times 10^{-3}$ | $6.82 \times 10^{-2}$ | 3 | | $7.202 \times 10^{-36}$ | $3.243 \times 10^{-35}$ | $5.417 \times 10^{-87}$ | $1.000 \times 10^{0}$ | $4.510 \times 10^{-58}$ |
| 10 | $6.30 \times 10^{-3}$ | $5.86 \times 10^{-2}$ | 3 | | $3.311 \times 10^{-28}$ | $6.318 \times 10^{-30}$ | $2.282 \times 10^{-61}$ | $1.000 \times 10^{0}$ | $1.830 \times 10^{-42}$ |
| 11 | $6.97 \times 10^{-3}$ | $5.96 \times 10^{-2}$ | 3 | | $2.972 \times 10^{-29}$ | $1.191 \times 10^{-30}$ | $1.055 \times 10^{-63}$ | $1.000 \times 10^{0}$ | $1.461 \times 10^{-43}$ |
| 12 | $-2.66 \times 10^{-2}$ | $-1.18 \times 10^{-2}$ | 4 | | $2.727 \times 10^{-28}$ | $2.310 \times 10^{-17}$ | $1.384 \times 10^{-65}$ | $1.521 \times 10^{-56}$ | $1.000 \times 10^{0}$ |
| 13 | $-2.68 \times 10^{-2}$ | $-1.64 \times 10^{-2}$ | 4 | | $8.075 \times 10^{-32}$ | $2.380 \times 10^{-17}$ | $1.771 \times 10^{-72}$ | $2.738 \times 10^{-59}$ | $1.000 \times 10^{0}$ |
| 14 | $-3.14 \times 10^{-2}$ | $-1.81 \times 10^{-2}$ | 4 | | $3.329 \times 10^{-36}$ | $3.051 \times 10^{-21}$ | $3.378 \times 10^{-87}$ | $2.106 \times 10^{-71}$ | $1.000 \times 10^{0}$ |
| 15 | $2.79 \times 10^{-2}$ | $2.76 \times 10^{-3}$ | 5 | | $9.995 \times 10^{-1}$ | $2.116 \times 10^{-6}$ | $5.257 \times 10^{-4}$ | $6.300 \times 10^{-19}$ | $3.457 \times 10^{-21}$ |
| 16 | $3.32 \times 10^{-2}$ | $2.46 \times 10^{-3}$ | 5 | | $1.000 \times 10^{0}$ | $2.820 \times 10^{-9}$ | $3.713 \times 10^{-9}$ | $7.638 \times 10^{-27}$ | $2.355 \times 10^{-27}$ |
| 17 | $2.41 \times 10^{-2}$ | $-1.54 \times 10^{-3}$ | 5 | | $9.779 \times 10^{-1}$ | $4.606 \times 10^{-3}$ | $1.751 \times 10^{-2}$ | $2.796 \times 10^{-15}$ | $1.285 \times 10^{-18}$ |
| 18 | $9.74 \times 10^{-3}$ | $-4.05 \times 10^{-3}$ | 6 | | $4.675 \times 10^{-5}$ | $9.986 \times 10^{-1}$ | $1.350 \times 10^{-3}$ | $2.120 \times 10^{-9}$ | $2.322 \times 10^{-8}$ |
| 19 | $1.57 \times 10^{-2}$ | $-2.33 \times 10^{-3}$ | 6 | Unknown | $1.208 \times 10^{-2}$ | $4.648 \times 10^{-1}$ | $5.231 \times 10^{-1}$ | $9.067 \times 10^{-10}$ | $1.078 \times 10^{-11}$ |
| 20 | $1.01 \times 10^{-2}$ | $-7.46 \times 10^{-4}$ | 6 | | $9.939 \times 10^{-4}$ | $5.405 \times 10^{-1}$ | $4.585 \times 10^{-1}$ | $7.914 \times 10^{-8}$ | $2.899 \times 10^{-7}$ |
| 21 | $4.78 \times 10^{-4}$ | $2.02 \times 10^{-2}$ | 7 | | $5.100 \times 10^{-2}$ | $1.262 \times 10^{-4}$ | $1.164 \times 10^{-2}$ | $9.322 \times 10^{-1}$ | $5.056 \times 10^{-3}$ |
| 22 | $1.33 \times 10^{-3}$ | $1.69 \times 10^{-2}$ | 7 | | $1.022 \times 10^{-1}$ | $1.459 \times 10^{-3}$ | $6.103 \times 10^{-1}$ | $2.326 \times 10^{-1}$ | $5.334 \times 10^{-2}$ |
| 23 | $2.57 \times 10^{-3}$ | $2.45 \times 10^{-2}$ | 7 | | $2.779 \times 10^{-4}$ | $8.184 \times 10^{-8}$ | $1.406 \times 10^{-5}$ | $9.997 \times 10^{-1}$ | $4.287 \times 10^{-7}$ |

**Known** 5 Groups

[0,1,2,16] [3,4,5] [6,7,8] [9,10,11] [12,13,14]

**Unknown** 1 Group

[15,17,18,19,20,21,22,23]

**Figure 8.** Instances grouped according to GMM probabilities and the threshold.

As previously mentioned, the GMM results were the first phase of the strategy and only served as a partial guide in the recognition of objects of unknown classes. In the second phase of the strategy, a clustering algorithm called MOCK was used, which was enhanced with NSGA-II. For the second phase, the 24 feature vectors without projection (288 features) were entered into the clustering algorithm. For execution, the algorithm was run with the parameters shown in Table 5.

**Table 5.** Parameters for the clustering algorithm MOCK/NSGA-II.

| Parameter | Values |
| --- | --- |
| Population size (M) | 9 |
| Nearest neighbors (L) | 2 |
| Number of generations | 10 |

The nine final individuals of the clustering process are shown in Figure 9 in terms of their fitness values, and Figure 10 shows how the vectors were grouped in different structures.

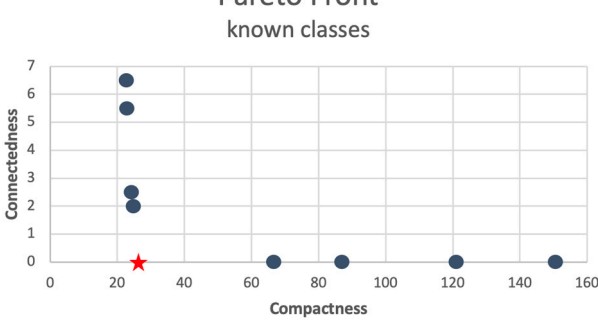

**Figure 9.** The Pareto front generated using the MOCK clustering algorithm improved with NSGA-II. The individual marked with a red star is the knee point.

```
[[0, 2, 1], [10, 9, 11], [12, 13, 14], [16], [17, 15], [18, 20], [19], [21, 22, 23], [3, 4, 5], [6, 8, 7]]
[[0, 2, 1], [10, 9, 11], [12, 13, 14], [16, 15], [17], [18, 20], [19], [21, 22, 23], [3, 4, 5], [8, 6, 7]]
[[0, 2, 1], [10, 9, 11], [12, 13, 14], [16, 15], [17], [18, 19, 20], [21, 22, 23], [3, 4, 5], [7, 8, 6]]
[[0, 2, 1], [10, 9, 11], [12], [13, 14], [16, 17, 15], [18, 19, 20], [21, 22, 23], [3, 4, 5], [8, 6, 7]]
[[0, 1, 2], [10, 9, 11], [12, 13, 14], [16, 17, 15], [18, 19, 20], [21, 22, 23], [3, 4, 5], [8, 6, 7]]
[[0, 2, 1], [10, 23, 21, 9, 11, 22], [12, 13, 14], [16, 17, 15, 19, 18, 3, 20, 4, 5], [8, 6, 7]]
[[0, 2, 1, 17, 16, 15], [10, 23, 21, 9, 11, 22, 12, 13, 14], [18, 19, 20], [3, 4, 5], [7, 8, 6]]
[[0, 2, 1, 17, 16, 19, 15, 18, 3, 20, 4, 5, 8, 6, 7, 21, 22, 23, 10, 9, 11], [12, 13, 14]]
[[0, 2, 1, 17, 16, 19, 15, 18, 3, 20, 4, 5, 8, 6, 7, 21, 22, 23, 12, 10, 13, 9, 11, 14]]
```

**9 Final Individuals**

**Figure 10.** The final nine individuals generated using the MOCK clustering algorithm improved with NSGA-II. The individual marked in red is the knee point.

Subsequently, the comparison described in Section 3.4 was performed to select the "best solution". The individuals generated using the MOCK/NSGA-II algorithm (Figure 10) and the known class subgroups generated using the GMM (Figure 8) were compared. The results of this comparison are shown in Table 6. It can be seen that Solutions 1, 2, 3, and 5 have four structures shared with the subgroups of known classes generated with the GMM. However, since the solution closest to the knee point was selected as the "best solution",

Solution 5 was chosen (marked with *), which in this case, was found to be the knee point, marked in red in Figure 10.

**Table 6.** Scores obtained from the comparison of the individuals generated using MOCK/NSGA-II and the known class subgroups generated using GMM. The solution marked with * was selected as the "best solution".

|  | Pareto Frontier Position | Score |
| --- | --- | --- |
| Solution 1 | 1 | 4 |
| Solution 2 | 2 | 4 |
| Solution 3 | 3 | 4 |
| Solution 4 | 4 | 3 |
| **Solution 5 *** | **5** | **4** |
| Solution 6 | 6 | 2 |
| Solution 7 | 7 | 2 |
| Solution 8 | 8 | 1 |
| Solution 9 | 9 | 0 |

Given the "best solution", the four clusters with shared structures with the known class subgroups generated with the GMM were determined as "*known classes*". Then, the clusters containing only objects that the GMM determined as unknown, shown in Figure 8, were determined as "*new classes*". The result of these processes can be seen in Figure 11.

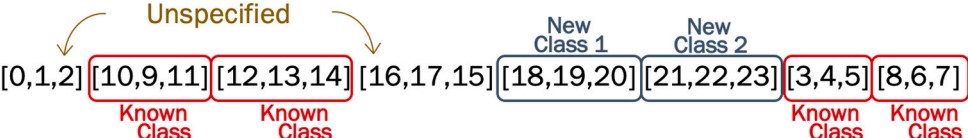

**Figure 11.** Selection of "*known classes*" and "*new classes*".

Since there were still unspecified clusters as *known* or *new*, we counted the number of known and unknown instances (as determined using the GMM) in the indeterminate clusters and defined the clusters in the same category as that comprising the majority of instances. Thus, we obtained five groups of "*known classes*" and three "*new classes*", as shown in Figure 12. Given the data in Table 4, we can confirm that indeed the vectors of the "new classes" corresponded to the instances of unknown objects and that they were grouped in the same structure as their "unknown class", thus confirming that both the "new classes" of objects of "unknown classes" can indeed be discovered.

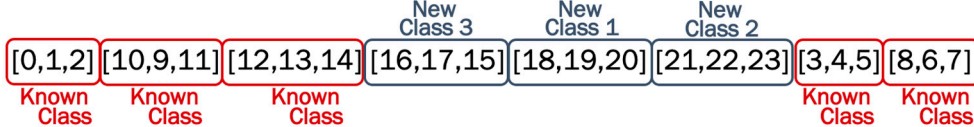

**Figure 12.** Final selection of "*known classes*" and "*new classes*".

Finally, the objects of known classes were entered into the classification section of the domain-specific CNN where a classification accuracy of 100% was obtained. Given the classification results obtained, we calculated the critical values of true positive (TP), false positive (FP), and false negative (FN) of both known and unknown classes. Subsequently, we calculated the micro-F1 score since it is one of the most commonly used metrics in OSR algorithms. The results obtained are shown in Table 7.

**Table 7.** Calculation of the micro-F1 score.

| | Label | True Positive (TP) | False Positive (FP) | False Negative (FN) | Micro-F1 |
|---|---|---|---|---|---|
| Known Classes | Chevrolet Silverado 2004 | 3 | 0 | 0 | |
| | Ford Explorer 2002 | 3 | 0 | 0 | Precision = 1.0 |
| | Ford Mustang 2000 | 3 | 0 | 0 | |
| | Honda Civic 2002 | 3 | 0 | 0 | |
| | Nissan Altima 2005 | 3 | 0 | 0 | Recall = 1.0 |
| Unknown Classes | Unknown Class 1 | 3 | 0 | 0 | |
| | Unknown Class 2 | 3 | 0 | 0 | |
| | Unknown Class 3 | 3 | 0 | 0 | Micro-F1 Score = 1.0 |
| | Total | 24 | 0 | 0 | |

## 5. Discussion and Conclusions

The main contribution of this work is to present a strategy to approach the VMMR as an OSR problem that is extended to the discovery of new classes, taking the distribution of feature vectors generated using a domain-specific CNN as the main guideline. This work seeks to highlight the importance of generating domain-specific OSR strategies and the need to apply them to real-world classification/recognition problems such as VMMR in order to obtain classifiers that are not only more accurate but also more robust, as they are prepared to face real-life scenarios. Although we focused on VMMR, the proposed methodology can be used as a benchmark for future domain-specific OSR problems and can be applied to other domains like handwritten digit recognition, chest X-ray classification, etc.

For the development of this work, we considered four main objectives to fulfill the purpose of approaching VMMR as an OSR problem extended for new class discovery. The fulfillment of our first objective could be validated with the results shown in Figure 6, where it can be seen that the CNN designed through the NE process with contrastive loss managed to map within the embedded space the feature vectors close in terms of cosine distance if they belonged to the same classes and far away if they belonged to different classes, maintaining this behavior for both known and unknown classes.

The second objective was described in detail in Section 3.4, which is the theoretical part of the third objective. In the Section 4, the proposed methodology was described step by step, and the experiments carried out validated that the proposed mechanism is able to detect objects of unknown classes and simultaneously discover their classes. One point to highlight is that our strategy is not restricted by training data, as it can be adjusted as these data change. More precisely, by using contrastive learning to train the feature extraction of the domain-specific CNN, the distribution of feature vectors is not only guided by "known classes" but is able to perform a consistent mapping even for objects of "unknown classes", which allows us to effectively detect objects of known classes and discover their classes simultaneously.

From the outset, we decided to employ a CNN not only to exploit the powerful ability of CNNs to extract meaningful features but also because these networks are known to be powerful classifiers. Therefore, since our domain-specific CNN was trained with numerous well-labeled examples, we could rely on its accuracy in classifying instances of known classes. Therefore, the last objective was met by achieving 100% classification accuracy of the images of the known classes in the test set.

Overall, the entire algorithm achieved a micro-F1 score of 1.00 by accurately classifying instances of known classes and effectively discovering the classes of instances whose classes were not included in the training. In a closed-set context, which is where most classification algorithms are developed, all instances of unknown classes would have been classified into some known class, so the model would not have been able to achieve a classification accuracy higher than 62.5% with the test set used in this work since 9 of the 24 test images belonged to unknown classes. The poor classification accuracy in this specific context, which simulates a real-life scenario, would be due to the incomplete knowledge of the world and not due to the classification potential that the classifier could achieve. Therefore, in this work, we proposed to add a mechanism to one of the most used image classifiers such as CNNs in order to detect objects of unknown classes and identify these classes. This highlights the possibility to expand the classification potential of CNNs and increase their robustness to work more effectively in real-life scenarios, thus enabling these classifiers not only to react to queries but also to continue learning even after being trained.

One of the limitations of this work was that due to time constraints, the neuroevolution algorithm was executed only seven times with the specified parameters, and it is left as future work to create a statistically more representative sample of executions and use a parameter calibration algorithm to possibly have better and more efficient results. It is also left as future work to increase the number of "known classes" to be able to classify more models with the domain-specific CNN and apply other OSR strategies to the VMMR problem for a more representative comparison.

**Author Contributions:** Conceptualization, D.-I.V.-S., H.-G.A.-M. and E.M.-M.; methodology, D.-I.V.-S., H.-G.A.-M. and E.M.-M.; software, D.-I.V.-S.; validation, H.-G.A.-M. and E.M.-M.; formal analysis, D.-I.V.-S.; investigation, D.-I.V.-S.; resources, D.-I.V.-S., H.-G.A.-M. and E.M.-M.; data curation, D.-I.V.-S.; writing—original draft preparation, D.-I.V.-S.; writing—review and editing, D.-I.V.-S., H.-G.A.-M. and E.M.-M.; visualization, D.-I.V.-S.; supervision, H.-G.A.-M. and E.M.-M. All authors have read and agreed to the published version of the manuscript.

**Funding:** This research received no external funding.

**Acknowledgments:** The first author would like to thank the Consejo Nacional de Ciencia y Tecnología (CONACYT), an institution of the Government of Mexico, for the financial support provided through the "Beca Nacional" with CVU 1141251 as part of the Programa de Becas para Estudios de Posgrado.

**Conflicts of Interest:** The authors declare no conflict of interest.

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
