# Peer review of "Vehicle Make and Model Recognition as an Open-Set Recognition Problem and New Class Discovery"

_mca, doi:10.3390/mca28040080_

Round 1

Reviewer 1 Report

Dear Authors

Some comments were highlighted in the attached pdf and should be addressed by the authors. Also, some sections of the manuscript must be improved for better understanding.

The novelty of the proposed framework is limited, combinations of deep neural networks and clustering techniques is not new. Even in the paper’s domain, Vehicle Make and Model Recognition, its contribution to accurately classify Make and Model for previously unseen Makes and Models doesn’t add must as these unseen cases are not classified but only clustered in different groups. Nevertheless, I’m willing to reconsider after major revision as it might be important for further studies on this topic to have these results for benchmarking. In general, all sections of the manuscript should be improved. Provide a complete revision in the next version.

The introduction section and discussion section, literature review should be more detailed and comprehensive. Authors should add more recent research. Notice that only 16 references were provided and only 7 were published between 2020-2022. Many of these references are not papers from web of science. References should be from the web of science 2020-2022 (50% of all references, 20 references at least).

Author Response

Reply to reviewers.

The authors thank the Editor and the reviewers for their helpful and pertinent comments and suggestions to our research. For helping in the reviewing process, the manuscript page numbers and line numbers where the corrections were done are included in each answer to facilitate their location.

Reviewer 1 Comments

Answer.

Comments highlighted in the attached pdf

not the best word in your context as standard IT definition is "Scalability is the measure of a system’s ability to increase or decrease in performance and cost in response to changes in application and system processing demands. Examples would include how well a hardware system performs when the number of users is increased, how well a database withstands growing numbers of queries, or how well an operating system performs on different classes of hardware. Enterprises that are growing rapidly should pay special attention to scalability when evaluating hardware and software."

Thanks for your comment. The term "scalability" was eliminated and instead the approach was shifted to an open set problem with a discovery of new classes.

i understand your point but stricly speaking this setence is incorrect, otherwise classifications algorithms would be worthless

Thanks for the observation. The explanation was adjusted to detail more precisely that although the classification algorithms are very powerful, most of them have the limitation of being developed in a closed set environment.

not clear, please re-write

Thanks for the observation. The adjustment in the approach was rewritten to provide better clarity in the explanation.

pls check previous related comment about scalability and change all text accordingly if you agree with my comment

We agree with the observation, the term "scalability" was removed.

impossible is too strong nowadays

Thanks for your comment. We change the expression to "very difficult".

add a paragraph about the technique "transfer learning" and its relation to this problem

Thank you for your suggestion. In the new version of the article we mention concepts such as incremental learning, transfer learning, zero-shot or one-shot (few-shot) learning, given the relation with some works, however, since we are not using any of these techniques we do not give an in-depth detail of them.

https://scholar.google.com/scholar?cites=15552448217139240756&as_sdt=2005&sciodt=0,5&hl=pt-PT

Unfortunately, this comment was not very clear to us. We add the GitHub link where the authors give access to their database.

please include a discussion section to compare your results with the other approaches

Thanks for your comment. Since the current VMMR approaches do not use OSR it was not possible to make a direct comparison, however a discussion of existing work and why our solution is more effective for the specific problem was added in the Introduction an Related work sections.

was it a a six category classification task? five + unknown

Thanks for the observation. In section 3.1 we gave a more detailed explanation of the number of classes and examples of each class.

pls define and clarify all variables and parameters in eq 1

Thanks for the observation. The eq.1 in the new version is eq.2 and a more complete detail of the variables was given.

explain this in more detail

Thanks for your comment. The modifications to the Neuroevolution algorithm and the justification for making them were explained in more detail in Section 3.2.

what fitness measure were used?

Thanks for your comment. A clearer explanation of the fitness function we use in the neuroevolution process was given in section 3.2.

include details about computer used

Thank you for your suggestion. In Section 4, the details of the environment where the process was executed were added.

include info loss details

Thank you for your suggestion. In Section 3.4, the detail of the percentage of variation of the PCA components used for the linear regression was given.

General review comments

The novelty of the proposed framework is limited, combinations of deep neural networks and clustering techniques is not new. Even in the paper’s domain, Vehicle Make and Model Recognition, its contribution to accurately classify Make and Model for previously unseen Makes and Models doesn’t add must as these unseen cases are not classified but only clustered in different groups. Nevertheless, I’m willing to reconsider after major revision as it might be important for further studies on this topic to have these results for benchmarking.

Thanks for your comment. The Framework within which the work was developed has been adjusted. There is a new field of research called Novel Class Discovery (NCD) where this problem is addressed with strategies similar to those we propose.

In general, all sections of the manuscript should be improved. Provide a complete revision in the next version.

The introduction section and discussion section, literature review should be more detailed and comprehensive.

Thanks for your comment. All sections of the manuscript have been modified to provide better detail.

Authors should add more recent research. Notice that only 16 references were provided and only 7 were published between 2020-2022. Many of these references are not papers from web of science. References should be from the web of science 2020-2022 (50% of all references, 20 references at least).

We especially thank you for this comment. A deeper investigation of the state of the art was carried out, which led to finding the correct term for the work we were doing. References were added as requested.

Author Response

Reply to reviewers.

The authors thank the Editor and the reviewers for their helpful and pertinent comments and suggestions to our research. For helping in the reviewing process, the manuscript page numbers and line numbers where the corrections were done are included in each answer to facilitate their location.

Reviewer 2 Comments

Answer.

The section “Literature review” is missing, please add options for the practical application of such methods in different industries, comparison and analysis of such methods, their shortcomings. Add literary sources, at least up to 25

Thank you for your suggestion. In the new version of the manuscript, Section 2 contains a review of the state of the art. Multiple references were added and commented as requested.

At the end of the section, clearly formulate the purpose of the study and the tasks of the work on the points: 1,2,3 ...

Unfortunately, this comment was not very clear to us. The proposed new approach and the purpose of this work was more clearly formulated.

The conclusions should have been made more specifically, in accordance with these tasks of the work (see paragraph 2). So far, they are more like a discussion.

Thanks for the observation. An adjustment was made to all sections including the conclusion section.

Make a separate section "Discussion and future research", there you can give examples of the legal application of car brand recognition in real objects (transport networks, parking lots, etc.) in the context of literary sources.

Thanks for the observation. The introduction section describes some of the services for which VMMR is functional.

Also write in which other industries (except cars) this technique can be applied

Thanks for your comment. Examples of other domains where this methodology could be applied were given in the conclusions section.

Make a graph or chart comparing 3-4 known methods and yours, which would show the advantages of your developments.

Thank you for your suggestion. Thanks for your comment. Since the current VMMR approaches do not use OSR it was not possible to make a direct comparison, however a discussion of existing work and why our solution is more effective for the specific problem was added in the Introduction and Related work sections.

Fig. 2 requires further explanation. Please sign the meaning of the lines of different colors on the graph (as I understand it, these are 7 starts of the process), why is one of the lines significantly lower?

Thanks for the observation. Each line represents one of the runs of the Neuroevolution algorithm. More clearly, each line represents the fitness that the best individual in each generation had. A more detailed explanation was given in Section 4 for better understanding.

As well as the meaning of the color of the lines and numbers of the code on the diagram (Fig.3). This scheme and the digital code under it also need to be described in more detail and analyzed.

Thanks for the observation. A more detailed explanation was given in order to understand the connections of the neuronal encoding.

Round 2

Reviewer 1 Report

Dear Authors,

Enhancements have been conducted and my comments have been taken into account. Please send a clean pdf without all those tracking notes. Eventually some minor revision might be needed, but I need that clean pdf to proceed.

Author Response

Thank you very much for your comment, we share the manuscript in pdf format as requested.

Reviewer 2 Report

This version of the article has improved its readability, the main comments have been corrected by the authors or explanations have been given.

There are also some unresolved issues, which are listed below.

1) The conclusions must be specified, give numerical indicators, write clearly point by point what has been done in accordance with the objectives of the study. Most of the conclusions that exist should be moved to the “discussion” section, including future research.

2) The objectives of the study also need to be formalized, described by points: 1,2,3 ... (I wrote about this in previous comments).

3) The reference list has been added, but now it needs to be sorted as it is mentioned. Also, in the review of similar studies, there are not enough references to authors who have previously dealt with this problem.

4) I think that a separate section with a literature review is needed (but I leave it to the discretion of the editors).

Author Response

Reply to reviewers.

The authors thank the Editor and the reviewer for their helpful and pertinent comments and suggestions to our research. For helping in the reviewing process, the manuscript page numbers and line numbers where the corrections were done are included in each answer to facilitate their location.

Reviewer 2 Comments

Answer.

The conclusions must be specified, give numerical indicators, write clearly point by point what has been done in accordance with the objectives of the study. Most of the conclusions that exist should be moved to the “discussion” section, including future research.

Thanks for your comment. The last section of the manuscript covers the discussion and conclusions of the work where the results of the work are analyzed and interpreted. Also, the results obtained are detailed point by point according to the objectives of the study.

The objectives of the study also need to be formalized, described by points: 1,2,3 ... (I wrote about this in previous comments).

Thanks for the observation. The specific objectives of the study were detailed at the beginning of section 3.

The reference list has been added, but now it needs to be sorted as it is mentioned. Also, in the review of similar studies, there are not enough references to authors who have previously dealt with this problem.

Thanks for your comment. The references were sorted as mentioned. In the introduction and literature review section, the 10 papers most related to ours are mentioned.

I think that a separate section with a literature review is needed (but I leave it to the discretion of the editors).

Thank you for your suggestion. Section 2 of the paper covers the literature review so the title was changed to "Literature Review".
